# Voting intentions during the later stage of the COVID-19 pandemic: The roles of risk perception and performance evaluations in South Korea

**Soo Yun Kim**[1], **Gyeongmin Kim**[2], **Seongoh Park**[3], **Cheonsoo Kim**[4], **Deok Hyun Jang**[5], **Dong-Hee Joe**[6‡*], **Won Mo Jang**[7,8‡*]

**1** Department of Communication, The University of Texas Rio Grande Valley, Edinburg, Texas, United States of America, **2** Department of Statistics, Sungshin Women's University, Seoul, Republic of Korea, **3** School of Mathematics, Statistics and Data Science, Sungshin Women's University, Seoul, Republic of Korea, **4** Division of Digital Media, Myongji University, Seoul, Republic of Korea, **5** Research Analytics & Communications, Gallup Korea, Seoul, Republic of Korea, **6** Department of Economics, Jeonbuk National University, Jeonju, Republic of Korea, **7** Department of Public Health and Community Medicine, Seoul Metropolitan Government–Seoul National University Boramae Medical Center, Seoul, Republic of Korea, **8** Department of Health Policy and Management, Seoul National University College of Medicine, Seoul, Republic of Korea

☯ These authors contributed equally to this work.
‡ These authors also contributed equally to this work.
* dongheejoe@jbnu.ac.kr (D-HJ), thomasj@snu.ac.kr (WMJ)

## Abstract

In this study, we examine voting intentions during the later stage of the COVID-19 pandemic in relation to risk perceptions and performance evaluations. Although prior research has documented rally-around-the-flag effects during the early phase of the pandemic, less is known about how vote choice is structured once acute crisis conditions have waned. This study draws on a nationally representative survey conducted in South Korea around the March 2022 presidential election. The analysis examines associations between affective and cognitive risk perceptions, evaluations of the government's COVID-19 countermeasures, overall government approval, economic evaluations and expectations, and voting intention. The results show that cognitive risk perception was associated with voting intention to a limited degree, whereas affective risk perception was not. More importantly, voting intentions were associated with conventional performance evaluations—overall government approval and economic evaluations and expectations—alongside pandemic-response approval, rather than being explained solely by the latter. This study addresses the need to better understand electoral behavior during the later phase of a prolonged crisis, a period that has received far less scholarly attention than the acute onset of the pandemic.

**Data availability statement:** The dataset supporting the conclusions of this article is publicly available through OpenICPSR at https://www.openicpsr.org/openicpsr/project/242181/version/V1/view.

**Funding:** The author(s) received no specific funding for this work.

**Competing interests:** DHJ is affiliated with Gallup Korea (https://www.gallup.co.kr/) but did not receive any funding from them for this work.

## Introduction

Political crises often reshape citizens' political attitudes and behavior by heightening perceptions of threat and uncertainty [1,2]. A large body of research shows that during crises, citizens tend to rally behind political leaders and incumbent governments, a phenomenon commonly referred to as the rally-around-the-flag effect [3,4]. This effect has been most consistently observed in response to security-related crises and public health emergencies, and is typically attributed to heightened threat perceptions, emotional responses such as fear and anxiety, and demands for decisive leadership [5,6].

However, recent scholarship suggests that rally-around-the-flag effects are often time-bound rather than permanent [7,8]. As crises persist, initial surges in trust and incumbent support may attenuate at the aggregate level, giving way to performance-based evaluations, economic concerns, and renewed political contestation [9]. In the context of the COVID-19 pandemic, research shows that as emergency conditions became normalized, aggregate rally-around-the-flag effects weakened [8]. Importantly, the attenuation of aggregate rally-around-the-flag effects does not imply that crisis-related perceptions become politically irrelevant. Research on COVID-19 demonstrates that perceived risk, uncertainty, and crisis-related anxiety often persist beyond the acute phase and continue to shape political attitudes and evaluations, even as objective health risks decline [10,11]. Recent studies further show that the political consequences of the pandemic have extended into the post-emergency period, influencing institutional trust, democratic attitudes, and policy preferences across different national contexts [12,13]. These findings raise new questions about whether and how underlying mechanisms such as risk perception and crisis-related government performance evaluations continue to influence political behavior during the later stages of prolonged crises, when uncertainty remains, even after the most acute phase of the crisis has passed.

South Korea provides an analytically informative case for examining how prolonged crises shape political behavior beyond the initial rally-around-the-flag phase because it experienced two nationwide elections at distinct stages of the same crisis. Research on crisis politics demonstrates that emotional responses to threat and uncertainty can influence electoral behavior across diverse democratic contexts, even when crises are triggered by different events and unfold under varying institutional arrangements [14,15]. In South Korea, the early phase of the COVID-19 pandemic coincided with the April 2020 National Assembly election, during which electoral outcomes were consistent with a pronounced rally-around-the-flag effect associated with the government's pandemic response [16]. Evidence from the April 2020 legislative elections further indicates that evaluations of the government's COVID-19 countermeasures became increasingly consequential for vote choice as the pandemic progressed.

By contrast, the later stage of the pandemic culminated in the March 2022 presidential election, held after crisis management had become routinized and acute emergency pressures had largely subsided. Analyses of this election emphasize the reassertion of conventional electoral mechanisms, including strong partisan

polarization, habitual partisanship, and retrospective evaluations of the incumbent administration, rather than crisis-driven rally-around-the-flag effects [17,18]. This temporal sequencing—an early election marked by rally-around-the-flag effects followed by a later election dominated by conventional electoral mechanisms—provides a suitable analytical context for examining how risk perceptions and evaluations of government performance (overall, pandemic response, and economic) relate to voting intentions as rally-around-the-flag effects attenuate over the course of a prolonged crisis.

Building on this context, we examine voter behavior in South Korea's March 2022 presidential election to assess how risk perceptions, evaluations of the government's handling of the COVID-19 pandemic, and economic assessments shaped voting intentions during the later stages of the prolonged crisis. Situating the analysis within the literature on crisis politics, this study moves beyond the identification of rally-around-the-flag effects to investigate how crisis-related perceptions and retrospective evaluations continue to operate once acute emergency pressures have subsided and electoral competition has resumed under routinized crisis management.

## Literature review

### Risk perception, government approval, and electoral behavior during the pandemic

Previous studies have conceptualized affective and cognitive risk perception as interrelated and mutually influential dimensions that shape individuals' evaluations of and responses to potential risks [19,20]. Cognitive risk perception reflects individuals' analytical assessments of the likelihood and potential severity of harm, grounded in deliberative reasoning about probabilities, outcomes, and available information [21]. In the context of the COVID-19 pandemic, this dimension is commonly operationalized through perceived likelihood of infection or expected health consequences and has been widely used to capture reasoned evaluations of personal vulnerability [22]. Affective risk perception, by contrast, captures individuals' emotional responses to perceived threats, such as fear, anxiety, or worry, which may arise independently of explicit probabilistic reasoning [23,24]. During the COVID-19 pandemic, affective risk perception was frequently measured through self-reported worry or emotional concern about infection and has been shown to play a central role in shaping protective behaviors, policy attitudes, and political responses to the crisis [25,26].

The dual-process theory has traditionally distinguished between affective and cognitive dimensions of risk perception, emphasizing emotionally versus analytically driven modes of evaluation [21,27]. At the same time, more recent work cautions against treating affective and cognitive risk perceptions as a strict dichotomy, instead conceptualizing them as interrelated and jointly predictive of judgment and choice [28,29]. For example, COVID-19 vaccination and other protective measures may have lowered perceived likelihood of infection while emotional responses such as worry persisted, reflecting the fact that these measures reduce but do not eliminate risk [22]. Accordingly, the present study examines how affective and cognitive risk perceptions were each associated with political attitudes during the later stage of the COVID-19 pandemic.

Extending these insights beyond individual health behavior [25,30–32], research has shown that risk perception also influences broader societal and political behaviors, including policy preferences, civic engagement, and voting behavior [33,34]. The Social Amplification of Risk Framework provides valuable insight into how risk perceptions, shaped and amplified through media coverage, institutional responses, and public discourse, can translate into political evaluations and electoral behavior during crises [35,36]. In the early stages of crisis, heightened risk perception is often associated with increased approval of incumbent governments and electoral support for political leaders [37,38], a phenomenon commonly described as the rally-around-the-flag effect [16]. In the COVID-19 context, fear of infection and the demand for decisive leadership initially contributed to elevated government approval, even when stringent measures restricted personal freedoms [38,39].

However, evidence also suggests that these rally-around-the-flag effects attenuate as crises become prolonged. As the pandemic progressed, economic hardship, policy fatigue, and perceived missteps in crisis management became more salient, leading to declining government approval and a re-emergence of conventional performance-based evaluations

[40,41]. The initially accepted restrictions and emergency measures became sources of frustration as citizens reassessed government effectiveness over time [38]. Despite growing evidence of such attenuation, it remains unclear whether and how risk perceptions continue to shape voter behavior in the later stages of prolonged crises. Therefore, the present study addresses this gap by examining the association between affective and cognitive risk perceptions and voting intentions during the late phase of the COVID-19 pandemic.

### Voter evaluation and retrospective voting in the COVID-19 pandemic

In usual (i.e., non-pandemic) elections, vote choices are well explained by retrospective voting, or post-election politics, which postulates that voters form expectations about the competence of the incumbent based on their evaluation of the incumbent's performance in office [42,43]. Voters punish "bad" performance by voting for the challenger and reward "good" performance by re-electing the incumbent, which induces the incumbent to demonstrate good performance in the first place [44]. In particular, this framework considers economic situation to be the most important performance measure (i.e., economic voting) [45]. Voters are more likely to support the incumbent when the economy is strong and turn to a challenger during economic downturns [46,47]. Extending this framework to elections held during the COVID-19 pandemic, the government's management of the pandemic was another important performance measure [48–50]. This suggests that we need to control for voters' economic evaluation and approval of the government, particularly of its handling of the pandemic. Our analysis thus includes voters' evaluations and expectations of the national economy and their own households, as well as voters' overall approval of the incumbent government and of its handling of the COVID-19 pandemic.

As reviewed earlier, existing studies report evidence that health concerns were major determinants of vote choices in elections during the early stages of the COVID-19 pandemic. As we include voters' economic evaluation and approval of the government in our analysis, known to be major factors in voters' decisions in non-pandemic elections, it would be interesting if health concerns remained stronger factors than voters' economic evaluation or government approval, or if the latter factors regained their prominence, in an election held during the end phase of the pandemic.

### Research questions and conceptual framework

To shed light on the association between risk perception and voting intention during the final phase of the COVID-19 pandemic, the following research questions (RQs) are addressed using survey data collected during the 2022 presidential election in South Korea:

RQ1. How did individuals' risk perceptions relate to their intention to vote for the incumbent party's candidate in an election held at the end of the COVID-19 pandemic?

RQ 2. What was the relationship between voters' evaluation of the government's performance and their intention to vote for the incumbent party's candidate in an election held at the end of the COVID-19 pandemic?

Fig 1 illustrates the study's conceptual framework, showing how political responses evolve across different phases of a prolonged crisis, from rally mobilization during the acute phase to pre-crisis political dynamics as crisis management becomes routinized. The framework also clarifies that both RQs are examined within a shared empirical context using data from a single cross-sectional survey.

## Materials and methods

### Study design

The study is based on a nationally representative cross-sectional survey of 1,001 South Korean voters conducted on March 2–3, 2022, approximately one week prior to the presidential election held on March 9, 2022. Although vaccination rates had increased by early 2022, the COVID-19 pandemic was not fully resolved at the time of data collection [51].

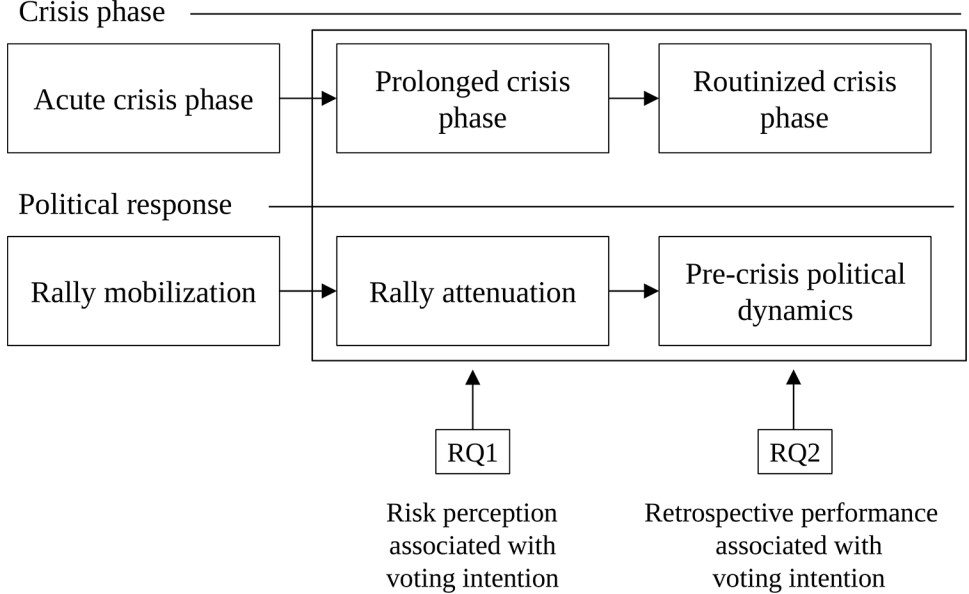

**Fig 1. Conceptual framework of crisis phases and political responses. Note: RQ1 = research question 1; RQ2 = research question 2.**

Breakthrough infections, the emergence of new variants (e.g., Omicron), and uncertainty regarding longer-term health consequences meant that COVID-19 continued to pose a salient public health concern, even though it was no longer in an acute phase [52]. Empirical evidence from South Korea further indicates that both cognitive and affective risk perceptions continued to fluctuate over the course of the pandemic rather than disappearing entirely [32].

## Sampling and weighting

The survey was administered by Gallup Korea, an affiliate of Gallup International, as part of a nationally representative cross-sectional election survey, via computer-assisted telephone interviews conducted by trained interviewers. Approximately 90% of interviews were conducted via mobile phones, and 10% via landlines using random-digit dialing. The sample was constructed using post-stratification by gender, age, and residential area to reflect the national electorate. To achieve national representativeness, we applied sampling weights in all analyses, including both descriptive statistics and logistic regression models. The total number of weighted responses used in the analysis was equal to the total number of unweighted cases at the national level. The weights were normalized to calculate the proportions and ratios, but not for estimating subtotal populations. The final sample analyzed in this study consisted of 658 respondents, after excluding respondents who had already been infected with COVID-19 (2%), those who selected neither approval nor disapproval on government approval measures (1.6%), and cases with missing data on key variables. Response options within individual questions were randomized by the survey organization, which helps mitigate potential response-order effects, although the order of questions themselves was not randomized.

## Measures

   **Outcome variable.** *Voting intention.* Voting intention was measured by asking respondents: "Which candidate do you plan to vote for in the upcoming presidential election?" Response options included the two major candidates (Yoon Seok-yeol and Lee Jae-myung), two minor candidates (Sim Sang-jung and Ahn Cheol-soo), "another candidate," or "none."

**Associated factors.** *Affective risk perception.* Respondents were asked, "How worried are you about getting infected with COVID-19?" on a 4-point Likert scale ranging from "Not worried at all" to "Very worried."

*Cognitive risk perception.* Respondents were asked, "How likely do you think you will be infected with COVID-19?" on a 4-point Likert scale ranging from "Not at all likely" to "Very likely."

*Approval of the incumbent government.* To capture overall evaluations of the incumbent government's performance, approval was measured by asking respondents whether they approved or disapproved of the incumbent government.

*Approval of the countermeasures to COVID-19 taken by the incumbent government.* To assess respondents' evaluations of the government's handling of the COVID-19 pandemic, approval of the government's countermeasures against COVID-19 was measured by asking respondents whether they approved or disapproved of the countermeasures implemented by the incumbent government.

*National economic evaluations.* To capture economic voting considerations, the survey measured both retrospective evaluations and prospective expectations of economic conditions at two levels: the national economy and respondents' own household economy. Retrospective national economic evaluation was measured by asking respondents to assess the national economy as improved, similar, or worsened over the preceding year. Prospective national economic expectation was measured by asking whether respondents expected the national economy to get better, remain the same, or get worse in the future.

*Household economic evaluations.* Retrospective household economic evaluation captured respondents' assessments of changes in their household's economic situation over the preceding year, with response options indicating whether conditions had improved, been similar, or worsened. Prospective household economic expectation measured respondents' expectations regarding their household's future economic situation using the same three-category responses (i.e., "It will get better," "It will remain the same," or "It will get worse").

*Control variables.* The survey also included socioeconomic characteristics, namely age, occupation, economic status, residential area, and educational attainment. Age was divided into five categories (18–29, 30–39, 40–49, 50–59, and 60 or above). Occupation was divided into seven categories (unemployed/retired/other, full-time homemaker, blue-collar worker, farming/forestry/fishery, white-collar worker, self-employed, and student). Economic status was divided into three categories (lower/lower middle, middle, and upper middle/upper). Residential area was divided into five categories: metropolitan (Seoul, Incheon and Gyeonggi Province), Yeongnam (south-eastern region), Chungcheong (middle region), Honam (south-western region), and none of the above (Gangwon and Jeju Province). Educational attainment was divided into three categories (middle school graduate or below, high school graduate, and college student or higher).

## Statistical analysis

Factors associated with COVID-19 risk perceptions were examined using a weighted logistic regression model. Descriptive comparisons of categorical variables were conducted using weighted chi-squared tests. All analyses incorporated survey weights to ensure national representativeness. Statistical significance was assessed with a significance level of 0.05.

## Ethical approval

This study was reviewed and approved by the Institutional Review Board of the Seoul Metropolitan Government-Seoul National University Boramae Medical Center (No. 20-2022-21). The institutional review board determined that the study posed minimal risk because responses were collected anonymously and were not linked to personally identifiable information. Accordingly, the requirement for obtaining informed consent was waived.

## Results

As shown in Table 1, slightly more than half of the respondents were men (52.3%), while white-collar workers (31.3%) and full-time homemakers (16.5%) collectively accounted for slightly less than half of the sample. Nearly half of the sample

**Table 1. Descriptive statistics and odds ratios between explanatory variables and voting intention.**

| Variables | Proportion | | | P-value (Chi-square test) | Adjusted odds ratio (95% confidence interval) | P-value |
|---|---|---|---|---|---|---|
| | **Voting intention** | | | | | |
| | Total | Suk-yeol Yoon | Jae-myung Lee | | | |
| (Intercept) | – | – | – | – | 0.01 (0.00-0.11) | <0.001§ |
| Gender | | | | 0.020* | | |
| Men | 52.3 | 28.4 | 23.9 | | 1.00 (Reference) | – |
| Women | 47.7 | 21.5 | 26.2 | | 2.67 (1.37–5.21) | 0.004‡ |
| Age | | | | <0.001*** | | |
| 18–29 | 16.4 | 9.5 | 6.9 | | 1.00 (Reference) | – |
| 30–39 | 14.2 | 5.2 | 9.0 | | 1.52 (0.49–4.69) | 0.467 |
| 40–49 | 21.1 | 7.8 | 13.3 | | 1.64 (0.51–5.23) | 0.404 |
| 50–59 | 21.3 | 9.3 | 12.0 | | 0.82 (0.26–2.59) | 0.732 |
| ≥ 60 | 27.0 | 18.2 | 8.8 | | 0.44 (0.13–1.48) | 0.184 |
| Occupation | | | | 0.018* | | |
| Unemployed/retired/other | 7.5 | 4.7 | 2.8 | | 1.00 (Reference) | – |
| Full-time homemaker | 16.5 | 9.5 | 7.1 | | 0.57 (0.14–2.34) | 0.438 |
| Blue-collar worker | 14.0 | 5.8 | 8.2 | | 1.40 (0.34–5.70) | 0.639 |
| Farming/forestry/fishery | 2.2 | 0.9 | 1.3 | | 1.40 (0.11–18.19) | 0.798 |
| White-collar worker | 31.3 | 13.4 | 17.9 | | 0.83 (0.24–2.89) | 0.771 |
| Self-employed | 20.0 | 10.6 | 9.5 | | 1.01 (0.26–3.88) | 0.992 |
| Student | 8.5 | 5.1 | 3.5 | | 0.13 (0.03–0.61) | 0.010‡ |
| Self-reported household economic status | | | | 0.067 | | |
| Lower/lower middle | 28.7 | 16.3 | 12.4 | | 1.00 (Reference) | – |
| Middle | 49.5 | 23.7 | 25.8 | | 1.26 (0.60–2.67) | 0.545 |
| Upper middle/upper | 21.8 | 9.9 | 11.9 | | 1.90 (0.78–4.62) | 0.158 |
| Residential area | | | | <0.001*** | | |
| Metropolitan area | 49.3 | 23.6 | 25.7 | | 1.00 (Reference) | – |
| Yeongnam | 25.1 | 16.1 | 9.1 | | 0.86 (0.42–1.77) | 0.686 |
| Chungcheong | 9.9 | 6.1 | 3.8 | | 0.57 (0.22–1.51) | 0.26 |
| Honam | 11.0 | 2.4 | 8.6 | | 6.09 (2.17–17.13) | <0.001§ |
| None of the above | 4.6 | 1.7 | 2.9 | | 0.92 (0.23–3.71) | 0.912 |
| Educational attainment | | | | 0.209 | | |
| Middle-school graduate or below | 7.3 | 3.9 | 3.4 | | 1.00 (Reference) | – |
| High-school graduate | 27.4 | 15.1 | 12.4 | | 0.76 (0.19–2.94) | 0.686 |
| College student or higher | 65.3 | 30.9 | 34.3 | | 0.89 (0.22–3.60) | 0.875 |
| Approval of the incumbent government | | | | <0.001*** | | |
| Disapproval | 51.9 | 45.4 | 6.5 | | 1.00 (Reference) | – |
| Approval | 48.1 | 4.5 | 43.6 | | 48.49 (21.99–106.93) | <0.001§ |
| Approval of the countermeasures to COVID-19 taken by the incumbent government | | | | <0.001*** | | |
| Disapproval | 45.7 | 38.6 | 7.2 | | 1.00 (Reference) | – |
| Neither disapproval nor approval | 4.6 | 1.8 | 2.8 | | 1.99 (0.62–6.38) | 0.248 |
| Approval | 49.7 | 9.5 | 40.2 | | 3.04 (1.46-6.32) | 0.003‡ |

*(Continued)*

**Table 1.** (Continued)

| Variables | Proportion | | | P-value (Chi-square test) | Adjusted odds ratio (95% confidence interval) | P-value |
|---|---|---|---|---|---|---|
| | Total | Voting intention | | | | |
| | | Suk-yeol Yoon | Jae-myung Lee | | | |
| Affective risk perception | | | | <0.001*** | | |
| Not worried at all | 14.8 | 7.6 | 7.1 | | 1.00 (Reference) | – |
| Not really worried | 26.0 | 10.4 | 15.7 | | 1.52 (0.55–4.21) | 0.419 |
| Somewhat worried | 37.8 | 18.3 | 19.5 | | 0.80 (0.30–2.12) | 0.656 |
| Very worried | 21.4 | 13.6 | 7.8 | | 0.90 (0.30–2.74) | 0.856 |
| Cognitive risk perception | | | | 0.079 | | |
| Not at all likely | 7.7 | 4.5 | 3.3 | | 1.00 (Reference) | – |
| Not really likely | 16.8 | 8.7 | 8.1 | | 2.55 (0.64–10.20) | 0.185 |
| Somewhat likely | 53.4 | 24.3 | 29.1 | | 3.16 (0.88–11.33) | 0.078 |
| Very likely | 22.1 | 12.5 | 9.6 | | 4.74 (1.11–20.31) | 0.036† |
| Retrospective national economic evaluation | | | | <0.001*** | | |
| Worsened | 52.4 | 39.2 | 13.2 | | 1.00 (Reference) | – |
| Similar | 32.3 | 9.4 | 22.9 | | 2.72 (1.36–5.44) | 0.005‡ |
| Improved | 15.3 | 1.3 | 14.0 | | 3.42 (1.10–10.61) | 0.033† |
| Prospective national economic expectation | | | | <0.001*** | | |
| It will get worse | 19.5 | 14.0 | 5.6 | | 1.00 (Reference) | – |
| It will remain the same | 41.4 | 21.0 | 20.4 | | 0.40 (0.18–0.88) | 0.024† |
| It will get better | 39.1 | 15.0 | 24.1 | | 0.36 (0.15–0.91) | 0.030† |
| Retrospective household economic evaluation | | | | <0.001*** | | |
| Worsened | 31.4 | 23.2 | 8.1 | | 1.00 (Reference) | – |
| Similar | 57.5 | 23.6 | 34.0 | | 1.02 (0.48–2.19) | 0.955 |
| Improved | 11.1 | 3.1 | 8.0 | | 3.41 (0.93–12.46) | 0.064 |
| Prospective household economic expectation | | | | <0.001*** | | |
| It will get worse | 10.4 | 8.9 | 1.5 | | 1.00 (Reference) | – |
| It will remain the same | 55.2 | 27.5 | 27.7 | | 4.22 (1.24–14.35) | 0.021† |
| It will get better | 34.3 | 13.5 | 20.8 | | 2.03 (0.55–7.46) | 0.289 |
| Voting intention | | | | – | | |
| Yoon Suk-yeol | 49.9 | – | – | | – | – |
| Lee Jae-myung | 50.1 | – | – | | – | – |

Chi-square tests (*p < .05; **p < .01; ***p < .001); logistic regression (†p < .05; ‡p < 01; §p < .001)

identified their household economic status as middle class (49.5%), and a similar proportion lived in a metropolitan area (49.3%). Most respondents reported a college education or higher (65.3%). Slightly less than half of the sample approved of the incumbent government's overall performance (48.1%), and approximately half of the sample approved of its handling of the COVID-19 pandemic. With respect to COVID-19 risk perception, more respondents were worried (21.4% very worried and 37.8% somewhat worried) than not worried (26.0% not really worried and 14.8% not worried at all), and expected to be infected (22.1% very likely and 53.4% somewhat likely) than not (16.8% not really likely and 7.7% not at all likely). Regarding economic conditions, a majority judged the national economy to have worsened over the preceding year

(52.4%), while most indicated that their household economic situation had remained similar (57.5%). Expectations for both national (41.4%) and household (55.2%) economic conditions were most commonly that conditions would remain the same.

Table 1 summarizes the results of the weighted chi-squared tests and the weighted logistic regression analyses addressing RQ1 and RQ2. The chi-squared tests examine the marginal associations between voting intention and each explanatory factor, indicating that voting intention differed significantly across most variables, with the exception of self-reported household economic status, educational attainment, and cognitive risk perception.

The logistic regression results are reported as adjusted odds ratios with 95% confidence intervals in parentheses. The reference category for the outcome variable (voting intention) is the main opposition candidate (i.e., Suk-yeol Yoon).

First, cognitive risk perception was positively correlated with voting intention for the incumbent party's candidate. Although it was not clear ex ante whether the rally-around-the-flag effect would persist in the end phase of the COVID-19 pandemic, this result is descriptively consistent with patterns discussed in the literature.

For government approval, both overall approval and countermeasures to the COVID-19 pandemic were positively correlated with intention to vote for the incumbent party's candidate. This finding was consistent with the aforementioned retrospective voting framework. Similarly, most variables related to the economic situation were positively correlated with intention to vote for the incumbent party's candidate. Specifically, those who judged the national economic situation as similar or improved were more likely to vote for the incumbent party's candidate than those who considered it to have worsened. Similarly, those who expected the national economy to remain the same or improve had a higher intention to vote for the incumbent party's candidate than those who expected it to worsen. Moreover, respondents who expected their households' economic situation to remain the same were more likely to vote for the incumbent party's candidate than those who expected it to worsen.

Some socioeconomic characteristics were also significantly correlated with voting intention. For instance, women were more likely to vote for the incumbent party's candidate, while students were less likely to vote for the incumbent candidate than those who were "unemployed/retired/other." Those living in Honam (southwest of the country) had higher intentions than those living in metropolitan areas.

## Discussion

This study uses survey data from the 2022 South Korean presidential election to examine how voting intentions during the later phase of the COVID-19 pandemic were associated with different dimensions of risk perception and broader evaluations of government and economic performance.

First, in the end phase of the COVID-19 crisis, cognitive risk perception showed a small but significant association with voting intention in the South Korean presidential election, whereas affective risk perception did not (RQ1). This contrasts with existing findings from earlier phases of the pandemic suggesting that affective risk plays a critical role in increasing the trust and approval of incumbent leaders, often triggering a rally-around-the-flag effect in response to heightened uncertainty [16,53]. The attenuated relationship between risk perception and voting intention observed in our study may reflect a broader decline in perceived risk as the pandemic persisted, and individuals adapted to the ongoing threat [54]. This is in line with prior research showing that prolonged exposure to crisis conditions can lead to desensitization or "risk fatigue," especially when individuals adapt to the threat and view it as manageable [20,36]. This pattern is also reflected in the 2023 Gallup Korea public opinion poll; while the overall approval of the Moon Jae-in administration's pandemic response remained relatively high, support for the ruling party's presidential candidate declined in the final months before the election [55]. In other words, voters' evaluations of COVID-19 risk and crisis management changed by the end of the pandemic.

One possible explanation, consistent with the literature on crisis normalization, is that everyday life under COVID-19 was increasingly perceived as routine during this period. South Korea's early and sustained success in containing outbreaks, transparent public health messages, and the institutionalization of protective behaviors (e.g., QR code check-ins, universal mask-wearing) may be interpreted as coinciding with a widespread perception that the crisis was under control

[56]. In the later phase of the pandemic, affective risk perception was less strongly associated with voters' evaluations of the incumbent and their responses to the pandemic. However, among the smaller group of voters reporting relatively high levels of attentiveness to infection trends and policy performance, cognitive risk perception was more salient in its association with voting preferences, particularly in support for the incumbent party, within a broader context of positive evaluations of South Korea's pandemic management. The Social Amplification of Risk Framework [35,36] suggests that risk perceptions are shaped not only by the objective characteristics of a threat but also by how institutions communicate and manage that threat over time. Even as COVID-19 became largely routine, continued exposure to official messaging, media framing, or personal experiences may help explain why perceived risk remained salient in some individuals' electoral decision-making [31,57,58].

Second, as the COVID-19 pandemic waned, voting intentions were associated not only with approval of the government's pandemic response but also with conventional performance evaluations, including overall government approval and economic evaluations and expectations (RQ2). These findings indicate that, during the later stage of a prolonged crisis, voting intentions were not structured primarily around pandemic-related approval but were instead associated with a broader set of performance-related considerations. This pattern can be situated within the broader literature on crisis politics. Prior research has consistently shown that rally-around-the-flag effects are inherently time-bound and tend to attenuate as crises become prolonged [7,8,40,41]. In the context of the COVID-19 pandemic, early electoral outcomes in multiple democratic settings—including South Korea's April 2020 National Assembly election—were widely interpreted as being consistent with rally-around-the-flag dynamics associated with approval of governments' pandemic responses [16]. These early patterns were linked to heightened uncertainty, perceived threat, and the political salience of crisis management during the acute phase of the pandemic.

In the South Korean context, pandemic governance was characterized by centralized coordination, extensive digital contact tracing, and sustained public communication, which contributed to the institutionalization of COVID-19 management as a routinized policy domain rather than an acute emergency [59–61]. Against this background, the associations observed in the present study are consistent with expectations derived from the crisis politics literature. Rather than suggesting that pandemic-related considerations disappeared from electoral judgment, the findings indicate that evaluations of COVID-19 countermeasures operated alongside other indicators of government performance in the later stage examined here. While the present study does not directly assess changes in rally-around-the-flag dynamics, this pattern is consistent with prior studies suggesting that the political salience of crisis-specific considerations diminishes over time as performance-based evaluations regain prominence [7,8,40,41].

These economic and general performance evaluations did not occur in isolation from the crisis [48,49,62]. Rather, they were observed alongside the broader consequences of prolonged public health restrictions, including lockdown-related economic disruptions, job insecurity, and fiscal relief efforts [63,64].

From this perspective, the present findings align with existing research demonstrating that evaluations of COVID-19 countermeasures retained political relevance beyond the acute phase of the pandemic [10–13,40,50,65]. Studies conducted across national contexts have shown that perceptions of pandemic response, perceived risk, and crisis-related anxiety continued to be associated with political attitudes and electoral preferences even as objective health risks declined [10–13,48,49]. In line with this literature, approval of the government's pandemic response remained significantly associated with voting intention in the present analysis, indicating that pandemic-related evaluations continued to form part of voters' political considerations.

Taken together, the findings indicate that during the later stage of the COVID-19 pandemic, voting intentions were associated with a configuration of performance evaluations that more closely resembles conventional retrospective patterns than crisis-centered ones. By focusing on the later phase of the pandemic, the present study extends previous work that has concentrated primarily on early crisis dynamics [16,40,41] and contributes to a more temporally differentiated understanding of how crisis-related and conventional evaluations coexist in shaping electoral behavior.

Finally, several limitations of this study should be noted. First, reliance on a pre-election survey without measures of voting intention strength limits assessment of how stated preferences translated into actual electoral behavior. Second, because the study is cross-sectional, the findings do not allow any causal interpretation or assessment of changes over time, underscoring the need for future longitudinal and qualitative research. Third, the distinction between cognitive and affective risk perception is subject to measurement limitations, as the survey design did not allow for sufficient ex ante consideration of design features such as randomization of question order. Finally, the findings should be interpreted in light of South Korea's institutional context, which offers a particularly informative case for understanding how electoral behavior unfolds during a prolonged public health crisis.

This study addresses the need to better understand electoral behavior during the later phase of a prolonged crisis, a period that has received far less scholarly attention than the acute onset of the COVID-19 pandemic. By examining how risk perceptions and performance evaluations relate to voting intentions at this stage, the study advances a more temporally differentiated perspective on crisis politics and electoral accountability. Academically, the study contributes by clarifying the distinct role of cognitive risk perception and by showing that evaluations of pandemic policies coexist with conventional assessments of government and economic performance in electoral judgment during a prolonged crisis. Practically, the findings imply that during extended pandemic conditions, governments cannot rely solely on early crisis management or public health achievements but must also demonstrate effective responses to ongoing economic and political challenges and communicate these efforts clearly to voters as active evaluators.

## Author contributions

**Conceptualization:** Soo Yun Kim, Seongoh Park, Cheonsoo Kim, Deok Hyun Jang, Dong-Hee Joe, Won Mo Jang.

**Data curation:** Deok Hyun Jang.

**Formal analysis:** Gyeongmin Kim, Seongoh Park.

**Investigation:** Deok Hyun Jang.

**Methodology:** Soo Yun Kim, Gyeongmin Kim, Seongoh Park, Deok Hyun Jang, Won Mo Jang.

**Supervision:** Dong-Hee Joe, Won Mo Jang.

**Validation:** Soo Yun Kim, Gyeongmin Kim, Seongoh Park, Cheonsoo Kim, Deok Hyun Jang, Dong-Hee Joe, Won Mo Jang.

**Visualization:** Gyeongmin Kim, Seongoh Park.

**Writing – original draft:** Soo Yun Kim, Gyeongmin Kim, Seongoh Park, Cheonsoo Kim, Deok Hyun Jang, Dong-Hee Joe, Won Mo Jang.

**Writing – review & editing:** Soo Yun Kim, Gyeongmin Kim, Seongoh Park, Cheonsoo Kim, Deok Hyun Jang, Dong-Hee Joe, Won Mo Jang.

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
