## [Decision Letter · Decision Letter 0]

6 Nov 2025

Dear Dr. Jang,

Thank you for submitting your manuscript to PLOS ONE. After careful consideration, we feel that it has merit but does not fully meet PLOS ONE’s publication criteria as it currently stands. Therefore, we invite you to submit a revised version of the manuscript that addresses the points raised during the review process.

We look forward to receiving your revised manuscript.

Kind regards,

Marco Improta

Academic Editor

PLOS ONE

“DHJ is affiliated with Gallup Korea (https://www.gallup.co.kr/) but did not receive any funding from them for this work.”

Reviewers' comments:

Reviewer's Responses to Questions

**Comments to the Author**

1. Is the manuscript technically sound, and do the data support the conclusions?

Reviewer #1: Partly

Reviewer #2: Yes

Reviewer #3: Yes

2. Has the statistical analysis been performed appropriately and rigorously?

Reviewer #1: No

Reviewer #2: Yes

Reviewer #3: Yes

3. Have the authors made all data underlying the findings in their manuscript fully available?

Reviewer #1: No

Reviewer #2: Yes

Reviewer #3: Yes

4. Is the manuscript presented in an intelligible fashion and written in standard English?

Reviewer #1: Yes

Reviewer #2: Yes

Reviewer #3: Yes

Reviewer #1: I commend the authors clarity of writing and identification of an extremely interesting and timely research question: how long does the rally around the flag (RATF) effect last?

The authors convincingly argue that the electoral and political events during the pandemic are an example of RATF, but they were at times followed by a sort of “backlash” or “boomerang” effect afterwards. So, how and why does the RATF effect end?

South Korea is a very convincing example, and the authors do a great work in presenting it as an appropriate case study. The writing is clear, although overall some sections could be cut to maintain focus on the main argument.

I think this work is very promising, but very early stages. It needs to undergo major revisions and possibly more data collection to make a meaningful contribution to this strand of literature, and I hope the authors will choose to move forward with this research.

Theory and data collection

This work is very interesting and promising. But at this stage, I’m afraid, your empirical and theoretical argument is not sufficiently developed.

Your main argument revolves around the distinction between affective and cognitive risk. This is theoretically interesting – but the way you operationalise it is not very convincing.

First, has this been done before? If your choice to operationalise as worry v. likely has been adopted before, I think you should say it. If you are introducing a methodological innovation, then you need to be more convincing of its theoretical and empirical validity.

Second, why do you assume that likelihood does not impact worry? I could be more worried about something that is more likely to happen. In your case, however, I am afraid that worry also influences likelihood. If your questions were not randomised, the response on the affective response question might prime respondents on the cognitive questions (people hate to contradict themselves).

Hope has been recently theorised as a bi-dimensional concept at the crossroads between valence and likelihood (https://www.researchgate.net/publication/343086814_Hope_During_Conflict). I think you could argue something similar about worry.

Finally, in 2022 the pandemic was largely beyond us. Why would people still be worried about it? And likelihood levels might be low, especially depending on vaccination rates (can you say something about that?).

Two other (important) issues: the order of questions should be randomised in survey. And data collection should be pre-registered. If this did not happen, you should tell the reader why and what are the consequences of these choices for the results.

Finally, you should better engage with literature on dual process theory. Many theoretical and empirical accounts now invite to go beyond this dichotomy. You can disagree but better to acknowledge that the debate exists.

Results and discussion

You should be very clear that you are not making any causal claim. You say “estimates may be biased when interpreted as causal effects”, but this is not accurate. There is no casual effect.

The discussion, then, is confusing. You seem to conclude that the most important thing was dissatisfaction with real estate policy, but then why focus on risk perception and covid? In the next paragraph I suggest a way to make sense of this – but it requires more data collection.

Lastly, you make too many assumptions about the mechanisms. From the data you have you can infer very little. You speculate that “affective risk perception may have carried less weight” and “cognitive risk perception may have continued”. These are interesting ideas but you need more supporting evidence to your claims. Maybe you can think of a mixed-methods design and complement survey data with interviews?

Minor issue: I wonder if you can adopt a more standard template for the regression table? It’s better if it’s all in one page, and you should have at least another layer of asterisks at p<.01.

Structure and engagement with the literature

The first paragraph of the introduction is unnecessary. Try to keep your writing focused on what you are trying to say, cutting down on unnecessary context. To me, there is no harm in starting the article at line 34. You can then immediately introduce South Korea as your case study, but focus more on why it can be a representative case for other countries or broader contexts (external validity). You can take inspiration from recent literature:

Vasilopoulos, P., Marcus, G.E., Valentino, N.A. and Foucault, M. (2019), Fear, Anger, and Voting for the Far Right: Evidence From the November 13, 2015 Paris Terror Attacks. Political Psychology, 40: 679-704. https://doi.org/10.1111/pops.12513

Lytkina, E., & Reeskens, T. (2024). Rally Around the Government or a Populist Response? How Concerns About COVID-19 and Emotional Responses Relate to Institutional Trust and Support for Right-Wing Populism. American Behavioral Scientist, 69(4), 478-504. https://doi.org/10.1177/00027642241240418 (Original work published 2025)

In your literature review section, you should be more focussed on what is the argument you are making. Your main RQ is “How does risk perception influence RATF effect after major crises”. Focus on these strands of literature: risk perception/propensity, crises, RATF. There is literature on risk propensity and voting beyond SARF, perhaps have a look whether it’s helpful.

You can use the COVID pandemic as a case study, but I would not make your argument around that. I think you can make it broader to crises in general. Perhaps, since South Korea underwent a major political crisis in 2024/5, you can collect new data and see if and how different crises trigger different RATF effects. And/or you can use the housing crisis. I think this would make your paper stronger and more original, and its impact more meaningful.

You also spend too much time on voters evaluation. I think government’s approval is a controlling variable, but does not need to have a central role in your theoretical framework.

One last thing on case selection: Can you tell me if there are any peculiar characteristics of South Korean society which might affect external validity? Any specific social or political traditions, conventions, or cultural worldviews, that may limit your conclusions to SK only? Maybe not, but as a reader I am wondering whether the effects are driven by something else other than risk perception.

Reviewer #2: The manuscript titled “Risk perception, government approval, and support for the incumbent party towards the end of COVID-19” provides an insightful and timely contribution to understanding voter behavior during the concluding phase of a prolonged crisis. The study is well-organized, theoretically grounded, and empirically relevant, particularly in the context of South Korea’s 2022 presidential election. It usefully shifts attention from the early “rally-around-the-flag” period to the late-stage political effects of the COVID-19 pandemic.

Overall, the paper is clear and coherent, and the findings are interesting and policy-relevant. I recommend acceptance after minor revisions to address several conceptual and methodological issues outlined below.

A conceptual model figure summarizing hypothesized relationships among variables would enhance reader understanding.

Clarify in the methods section whether the weighting was applied in regression analysis or only for descriptive statistics.

Improve phrasing in the abstract’s conclusion—currently, the statement that “performance voting may outweigh the rally-around-the-flag effect” may overstate the evidence; softening this claim would improve precision.

Reviewer #3: he manuscript was written very clearly in English, which made it easy to follow. It presented significant and important new findings that were well-supported because all the data was transparent and simple to see. We could also easily understand the statistical analysis, making the whole paper exceptionally clear and its conclusions highly reliable....

**Do you want your identity to be public for this peer review?** For information about this choice, including consent withdrawal, please see our For information about this choice, including consent withdrawal, please see our Privacy Policy .

Reviewer #1: No

Reviewer #2: **Yes:** Muhammad Shakeel Ahmad SiddiquiMuhammad Shakeel Ahmad Siddiqui

Reviewer #3: **Yes:** Abdulhamid AbuanizaAbdulhamid Abuaniza

---

## [Author Response · Author response to Decision Letter 1]

9 Jan 2026

Response to peer reviewer comments

Dear Editor:

Thank you for giving us the opportunity to submit a revised version of our manuscript, “Voting intentions during the later stage of the COVID-19 pandemic: The roles of risk perception and performance evaluations in South Korea” (revised manuscript, PONE-D-25-43702), to PLOS ONE. We appreciate the time and effort you and the reviewers have dedicated to providing valuable feedback on the manuscript. We are also grateful to the reviewers for their insightful comments on the paper. We have been able to incorporate changes in response to a majority of the suggestions provided by the reviewers. We have included the line numbers of the changes within the response sheet.

Below is a point-by-point response to the reviewers’ comments and concerns.

Journal Requirement

Response: We have updated the manuscript in accordance with PLOS ONE’s style requirements.

2. The journal requires confirmation that the disclosed competing interest affiliation does not affect the authors’ adherence to PLOS ONE policies on data and materials sharing.

Response: We have updated the competing interest statement according to PLOS ONE policies.

3. The journal requires clarification that any restrictions on data sharing are based on legal or ethical grounds, as PLOS ONE permits data to be available upon request only under such conditions.

Response: In compliance with PLOS ONE policies, the data will be made publicly available at the following address: https://www.openicpsr.org/openicpsr/project/242181/version/V1/view The dataset is currently being prepared for public release and will be accessible immediately after the article is formally published.

Comments from Reviewer 1

1. General

I commend the authors clarity of writing and identification of an extremely interesting and timely research question: how long does the rally around the flag (RATF) effect last?

Response: Thank you for your positive evaluation of our research.

The authors convincingly argue that the electoral and political events during the pandemic are an example of RATF, but they were at times followed by a sort of “backlash” or “boomerang” effect afterwards. So, how and why does the RATF effect end?

Response: In the revised manuscript, we clarify that our focus is not on the presence or absence of a rally effect per se, but on documenting how different dimensions of risk perception and performance evaluation are associated with voting intentions in the later stage of a prolonged crisis (Discussion, lines 335–373).

Specifically, we examine whether affective and cognitive risk perceptions, alongside evaluations of government performance, remained associated with political behavior during the later phase of the COVID-19 pandemic, when acute emotional urgency had diminished and evaluative considerations were more salient (Introduction, lines 138–160).

2. Theory and data collection

This work is very interesting and promising. But at this stage, I’m afraid, your empirical and theoretical argument is not sufficiently developed. Your main argument revolves around the distinction between affective and cognitive risk. This is theoretically interesting – but the way you operationalise it is not very convincing. First, has this been done before? If your choice to operationalise as worry v. likely has been adopted before, I think you should say it. If you are introducing a methodological innovation, then you need to be more convincing of its theoretical and empirical validity.

Response: Thank you for encouraging us to clarify the theoretical and empirical grounding of our operationalization. In the revised manuscript, we now explicitly situate the distinction between affective and cognitive risk perception within established risk-perception research and clarify that this approach is not intended as a methodological innovation. Specifically, we note that prior research conceptualizes affective and cognitive risk perceptions as interrelated dimensions and that dual-process theory has long distinguished between emotionally and analytically driven modes of risk evaluation. We further clarify that, consistent with existing COVID-19 research, cognitive risk perception is commonly operationalized through perceived likelihood or expected health consequences, while affective risk perception is measured through self-reported worry or emotional concern (Introduction, lines 77–121). These revisions are intended to make clear that our operationalization follows established practice rather than introducing a novel measurement strategy.

Second, why do you assume that likelihood does not impact worry? I could be more worried about something that is more likely to happen. In your case, however, I am afraid that worry also influences likelihood. If your questions were not randomised, the response on the affective response question might prime respondents on the cognitive questions (people hate to contradict themselves). Hope has been recently theorised as a bi-dimensional concept at the crossroads between valence and likelihood (https://www.researchgate.net/publication/343086814_Hope_During_Conflict). I think you could argue something similar about worry.

Response: We appreciate your insightful connection to recent work on hope as a bi-dimensional construct integrating valence and perceived likelihood (Leshem & Halperin, 2020). We neither assume that perceived likelihood does not influence worry nor do we assume unidirectional causality in the opposite direction. Rather, consistent with prior research, we treat affective and cognitive risk perceptions as interrelated components that may mutually influence one another over time, while retaining an analytic distinction for measurement purposes (Introduction, lines 77–121).

We acknowledge your concern regarding potential priming effects arising from question order. However, our aim is not to disentangle causal directionality between worry and likelihood but to examine how each dimension is associated with political attitudes at a specific point in time.

Finally, in 2022 the pandemic was largely beyond us. Why would people still be worried about it? And likelihood levels might be low, especially depending on vaccination rates (can you say something about that?).

Response: While vaccination rates had increased by early 2022, the COVID-19 pandemic was not fully resolved at the time of data collection, as breakthrough infections, emerging variants, uncertainty about long-term health effects, and uneven risk exposure persisted. Accordingly, both affective and cognitive risk perceptions continued to vary meaningfully across individuals, even in the later phase of the pandemic. We have revised the first paragraph of the “Data collection” subsection of the Methodology and added additional citations (Materials and methods, lines 163–172).

Two other (important) issues: the order of questions should be randomised in survey. And data collection should be pre-registered. If this did not happen, you should tell the reader why and what are the consequences of these choices for the results.

Response: The data analyzed in this study were drawn from a nationally representative cross-sectional election survey conducted by Gallup Korea. The questionnaire design, including the order of questions, was finalized prior to data collection and did not allow for subsequent modification. While the order of questions was not randomized, response options within individual questions were randomized by the survey organization, helping to mitigate potential response-order effects. Although question-order effects cannot be entirely ruled out, all respondents were exposed to the same standardized question sequence, which reduces the likelihood of systematic bias in comparisons across groups. To clarify these points, we have revised the text to explicitly describe the survey’s administration by Gallup Korea as part of a nationally representative election survey using computer-assisted telephone interviews (Materials and methods, lines 174–189).

In addition, because the data were not preregistered and were collected as part of a time-sensitive election survey, the analyses are best interpreted as correlational rather than confirmatory. The findings are guided by established theory and consistent with prior research, but they do not support causal claims (Discussion, lines 383–385).

Finally, you should better engage with literature on dual process theory. Many theoretical and empirical accounts now invite to go beyond this dichotomy. You can disagree but better to acknowledge that the debate exists.

Response: We have revised the literature review to more explicitly engage with ongoing debates surrounding dual-process approaches to risk perception. We now clarify that while affective and cognitive risk perceptions are often distinguished in the literature, they are widely understood as interrelated rather than as a strict dichotomy. We further explain why prior research has commonly measured worry and perceived likelihood separately—particularly in contexts such as COVID-19, where protective measures may reduce perceived likelihood without fully alleviating emotional concern—and clarify that the present study follows this established practice. These revisions acknowledge the broader theoretical debate while clarifying our analytic approach (Introduction, lines 77–121).

3. Results and discussion

You should be very clear that you are not making any causal claim. You say “estimates may be biased when interpreted as causal effects”, but this is not accurate. There is no casual effect.

Response: We agree that, given the cross-sectional nature of our data, the findings should not be interpreted as causal. In response to your comment, we have revised the manuscript to remove language in the Discussion that could be misinterpreted as implying causality, including the paragraph in question. We hope that this revision clarifies our non-causal analytical stance and reduces any potential confusion for readers.

The discussion, then, is confusing. You seem to conclude that the most important thing was dissatisfaction with real estate policy, but then why focus on risk perception and covid? In the next paragraph I suggest a way to make sense of this – but it requires more data collection.

Response: We agree that the Discussion could be confusing, particularly regarding the emphasis on real estate-specific issues, which risked obscuring the core argument of the paper; accordingly, we have removed these references from the revised manuscript. Instead, we reinterpret the findings by emphasizing that conventional performance evaluations emerged as significant alongside assessments of pandemic response capacity, rather than being displaced by them. Although the cross-sectional design does not allow us to directly examine causal relationships or changes over time, we draw on existing literature on crisis politics and retrospective voting to offer an interpretive account of a transition from pandemic-dominated electoral dynamics to a mixed configuration in which pre-crisis performance considerations re-emerge while pandemic-related issues continue to coexist. To support this reframing, we have substantially revised the Introduction, conceptual framework (Figure 1), and Discussion. We hope these revisions address your concern and clarify the paper’s central contribution (Introduction, lines 21–160; Discussion, lines 299-380).

Lastly, you make too many assumptions about the mechanisms. From the data you have you can infer very little. You speculate that “affective risk perception may have carried less weight” and “cognitive risk perception may have continued”. These are interesting ideas but you need more supporting evidence to your claims. Maybe you can think of a mixed-methods design and complement survey data with interviews?

Response: We agree that the original discussion risked being read as overinterpreting underlying mechanisms that cannot be directly identified with cross-sectional survey data. In the revised manuscript, we have therefore taken care to avoid definitive or mechanistic claims, while retaining an explicitly interpretive and theory-informed discussion of possible explanations that is grounded in and consistent with prior literature. In particular, statements regarding affective risk perception carrying less weight and cognitive risk perception remaining salient are now framed as plausible interpretations rather than empirical demonstrations, informed by existing research on crisis normalization and risk fatigue (Introduction, lines 77–122; Discussion, lines 303–334). At the same time, we explicitly acknowledge that our design does not allow direct examination of mechanisms or temporal change, and we note that mixed-methods approaches, including qualitative interviews, would be necessary to address these questions in future research (Limitations, lines 383–385). We hope these revisions clarify the scope of our claims and directly address concerns about overinterpretation while preserving the analytical value of theory-informed interpretation.

Minor issue: I wonder if you can adopt a more standard template for the regression table? It’s better if it’s all in one page, and you should have at least another layer of asterisks at p<.01.

Response: In line with common practice in published empirical studies, we have merged the summary statistics and odds ratio results into a single table (Table 1, line 265) to improve clarity and ensure that the regression results are presented on one page.

Moreover, we have added an additional layer of significance indicators to distinguish between p < .01, p < .05, and p < .10, and we have also included exact p-values in the table to facilitate interpretation and transparency (see the note below Table 1).

4. Structure and engagement with the literature

The first paragraph of the introduction is unnecessary. Try to keep your writing focused on what you are trying to say, cutting down on unnecessary context. To me, there is no harm in starting the article at line 34. You can then immediately introduce South Korea as your case study, but focus more on why it can be a representative case for other countries or broader contexts (external validity). You can take inspiration from recent literature:

Vasilopoulos, P., Marcus, G.E., Valentino, N.A. and Foucault, M. (2019), Fear, Anger, and Voting for the Far Right: Evidence From the November 13, 2015 Paris Terror Attacks. Political Psychology, 40: 679-704. https://doi.org/10.1111/pops.12513

Lytkina, E., & Reeskens, T. (2024). Rally Around the Government or a Populist Response? How Concerns About COVID-19 and Emotional Responses Relate to Institutional Trust and Support for Right-Wing Populism. American Behavioral Scientist, 69(4), 478-504. https://doi.org/10.1177/00027642241240418 (Original work published 2025)

Response: Following your comment, we substantially revised and restructured the Introduction to improve focus and theoretical clarity, rather than retaining a broad descriptive overview of the pandemic (Introduction, lines 21–74). The revised manuscript now opens directly with the literature on crisis politics, rally-around-the-flag dynamics, and affective versus cognitive mechanisms. South Korea is introduced early as the empirical case, and the revised Introduction explicitly explains that the country offers a particularly informative analytical setting because nationwide elections were held both during the early and later phases of the COVID-19 pandemic, allowing examination of voting intentions as pandemic-related risks became normalized rather than acute (Introduction, primarily lines 44–74). In addition, the Discussion has been revised to situate the findings within cross-national literature on crisis politics by highlighting how risk perception and retrospective performance evaluations can coexist during the later phase of a prolonged crisis, rather than assuming crisis-driven voting dynamics limited to the acute phase (lines 335–380). We have also incorporated the suggested references into the Introduction and added them to the

---

## [Decision Letter · Decision Letter 1]

9 Mar 2026

Voting intentions during the later stage of the COVID-19 pandemic: The roles of risk perception and performance evaluations in South Korea

PONE-D-25-43702R1

Dear Dr. Jang,

We’re pleased to inform you that your manuscript has been judged scientifically suitable for publication and will be formally accepted for publication once it meets all outstanding technical requirements.

Kind regards,

Ching-Hsing Wang, Ph.D.

Academic Editor

PLOS One

Additional Editor Comments (optional):

Reviewers' comments:

Reviewer's Responses to Questions

**Comments to the Author**

Reviewer #1: All comments have been addressed

Reviewer #3: All comments have been addressed

2. Is the manuscript technically sound, and do the data support the conclusions?

Reviewer #1: Yes

Reviewer #3: Yes

3. Has the statistical analysis been performed appropriately and rigorously?

Reviewer #1: Yes

Reviewer #3: Yes

4. Have the authors made all data underlying the findings in their manuscript fully available?

Reviewer #1: Yes

Reviewer #3: Yes

5. Is the manuscript presented in an intelligible fashion and written in standard English?

Reviewer #1: Yes

Reviewer #3: Yes

Reviewer #1: Thank you all for thoroughly engaging with my earlier comments and for the productive discussion which I think has improved the paper.

Reviewer #3: The author has addressed all the comments and the points in a clear manner. The manuscript presented in very good and standard English.

**Do you want your identity to be public for this peer review?** For information about this choice, including consent withdrawal, please see our For information about this choice, including consent withdrawal, please see our Privacy Policy .

Reviewer #1: No

Reviewer #3: **Yes:** Abdulhamid AbuanizaAbdulhamid Abuaniza

---

## [Editor Report · Acceptance letter]

PONE-D-25-43702R1

PLOS One

Dear Dr. Jang,

I'm pleased to inform you that your manuscript has been deemed suitable for publication in PLOS One. Congratulations! Your manuscript is now being handed over to our production team.

Kind regards,

on behalf of

Associate professor Ching-Hsing Wang

Academic Editor

PLOS One